# Effect of Neutral Protease on Freshness Quality of Shucked Pacific Oysters at Different Storage Conditions

**DOI:** 10.3390/foods13081273

**Published:** 2024-04-21

**Authors:** Lanxiang Su, Wenge Yang, Siyang Liu, Chunhong Yuan, Tao Huang, Ru Jia, Huamao Wei

**Affiliations:** 1College of Food Science and Engineering, Ningbo University, Ningbo 315211, China; 2Key Laboratory of Animal Protein Food Deep Processing Technology of Zhejiang Province, Ningbo University, Ningbo 315211, China; 3Faculty of Agriculture, Iwate University, Ueda 3-18-8, Morioka 020-8550, Iwate, Japan

**Keywords:** *Crassostrea gigas*, neutral protease, freshness, AEC, storage, superchilling

## Abstract

The aim of this study was to investigate the effect of neutral protease treatment on the biochemical properties of various parts of Pacific oysters (*Crassostrea gigas*) under different storage conditions. The mechanism of quality degradation in the mantle, adductor muscle, gill, and trunk of treated oysters stored at −1.5 °C (superchilling) or 4 °C (refrigeration) for several days using different storage methods was studied. The results showed that the oyster treated with the enzyme exhibited higher glycogen content, flavor nucleotide content, and sensory scores compared to the control group. Superchilling at −1.5 °C was observed to slow the increase in total volatile basic nitrogen (TVB-N), total viable count (TVC), and pH, while maintaining sensory scores better than refrigeration at 4 °C. Both wet superchilling (WS) and dry exposed superchilling (DeS) methods effectively preserved freshness and quality at −1.5 °C. The freshness of the oysters’ body trunk changed most significantly. K value, K′ value, and AEC value, as the evaluation indexes of oyster freshness, were affected by the storage medium. Therefore, neutral protease enhances the flavor of oysters in a short time, and oysters stored in wet superchilling or dry exposed superchilling conditions have an extended shelf life.

## 1. Introduction

The Pacific oyster (*Crassostrea gigas*) is a popular type of bivalve shellfish in China, known for its high-quality taste and freshness, which significantly influences its market value [1]. As with most aquatic products, the high moisture content and high endogenous enzyme activity of oysters were the main factors contributing to their quality deterioration during storage, along with the growth of microorganisms [2]. Many surveys have indicated that most consumers prefer shucked oysters (pretreated oysters) over unshucked oysters due to the perception that handling seafood is time-consuming and requires specific knowledge [3]. Consumers also consider the flavor and freshness of shucked oysters when making their choices. Therefore, it is necessary to explore a method to enhance the flavor of shucked oysters and investigate the quality degradation mechanism during storage. Currently, many studies have confirmed that the addition of exogenous proteases could improve the flavor quality of aquatic products in a short time by promoting the generation of flavor amino acids and flavor nucleotides [4,5]. While most of the present studies have focused on exploring the production of umami peptides from homogenized oysters using hydrolytic enzyme treatment [6,7], this method may not align with typical consumption habits. In addition, using flavor enzymes to treat oysters has inevitably reduced their shelf life by degrading proteins and promoting microbial growth. Studies related to the shelf life of oysters treated with hydrolytic enzymes are scarce. Therefore, developing effective storage methods for enzyme-treated shucked oysters would be beneficial.

Up to now, low-temperature preservation has been considered an effective method to extend the shelf life of aquatic products, which could be classified into frozen preservation [8], superchilling preservation [9], and refrigeration, based on specific temperature range. Although frozen storage provides a significant extension of the shelf life in aquatic products, the thawing step before consumption might cause significant nutrient loss and flavor deterioration [10]. Superchilling is a method that reduces the temperature of the food product to 1–2 °C below the initial freezing points to extend the shelf life by inhibiting microbial and enzymatic activities [11]. However, its effectiveness in prolonging shelf life is still controversial because temperature fluctuations may easily contribute to the formation of ice crystals and the occurrence of the freeze–thaw phenomenon [12]. Whereas the shelf life of aquatic products stored in refrigeration is usually short, for oysters, survival storage could be used for long-term preservation, maintaining the quality of the oysters. Therefore, it is important to find ways to improve the flavor of oyster meat in a short time, develop cryopreservation methods, and predict the shelf life of oysters for the oyster market to align with the current consumer demand for shucked oysters. In addition, studies on storage methods for oysters, such as wet storage, dry exposed storage, and dry storage, and their impact on the biochemical properties and shelf life of oysters remains limited.

Therefore, to clarify the effect of storage method and temperature on the quality of shucked oysters treated with flavourzyme, various parameters, including glycogen content, pH value, volatile basic nitrogen (TVB-N) value, total viable counts (TVC), and ATP-related compounds, were measured in different parts of the oyster to investigate the mechanisms of quality deterioration and predict the shelf life of oysters.

## 2. Materials and Methods

### 2.1. Materials

Live Pacific oysters (*Crassostrea gigas*), weighing (123.24 ± 9.74 g), were purchased from the local aquatic market. All oysters were placed in sterilized seawater for more than 2 h, and only those with shells opened less than 3 mm were selected as samples for this study.

Firstly, all oysters were shucked and cleaned with precooled sterilized saline, and surface excess water was removed using filter paper. The control group was stored after being immersed in ice water for 1.5 h, while the experimental groups were further divided into 4 groups following a 1.5 h immersion in 0.5% neutral protease (*w*/*v* 1:2) at 0 °C. The whole bodies of the WS (wet superchilling) and WR (wet refrigeration) groups were packed in sterilized PTFE bags containing 3.5% (*w*/*v*) NaCl solution. The samples of the DS (dry superchilling) group were placed directly into sterilized PTFE bags, while the samples wrapped in wet cloths soaked in 3.5% NaCl solution were classified as the DeS (dry exposed superchilling) group. The WS, DS, and DeS groups were stored at −1.5 ± 0.2 °C for up to 13 days, while the WR and control groups were placed at 4.0 ± 0.2 °C for 7 days until unpleasure flavor appeared. To better investigate the effects of storage method and temperature on oyster quality, oysters were divided into four parts: mantle, adductor muscle, gill, and other parts (body trunk), and the changes in TVB-N, glycogen, and ATP-related compounds of each part during storage were determined. In addition, whole oyster TVB-N values, TVC, and sensory evaluation were also used to estimate oyster quality.

Seven nucleotide-related compound standards (ATP, ADP, AMP, IMP, AdR, Hx, HxR), all of which were chromatographically pure, were purchased from Shanghai Amperex Scientific Instrument Co., Ltd. (Shanghai, China). The other materials included sodium hydroxide, sodium chloride, hydrochloric acid, boric acid, magnesium oxide, ethanol, methyl red, bromocresol green, perchloric acid, sodium dihydrogen phosphate, disodium hydrogen phosphate, and phosphoric acid (analytical pure, Sinopharm Chemical Reagent Co., Ltd., Shanghai, China). Agar culture medium (analytical pure) was obtained from Hangzhou Microbial Reagent Co., Ltd., Hangzhou, China.

### 2.2. Determination of Total Volatile Basic Nitrogen (TVB-N)

The TVB-N of the samples was determined as described previously [2]. A 2 g sample was homogenized with 10 times the volume of water using a homogenizer (XHF-DY, Scientz, Ningbo, China) at 10,000 rpm for 1 min and then centrifuged using a high-speed freezing centrifuge (Centrifuge 5804 R, Eppendorf, Germany) at 4000× *g* for 10 min at 4 °C. Subsequently, 5 mL of supernatant and 5 mL of 10 g/L MgO solution were added to a Kjeldahl tube, and the TVB-N was measured using a Kjeldahl nitrogen analyzer (K9840, Sartorius Scientific Instruments, Beijing, Co., Ltd., Beijing, China). The TVB-N values were calculated as follows:*TVB-N* (mg/100 g) = [(*V*_1_ − *V*_2_) × *C* × 14]/[*M* × (10/100)] × 100(1)
where *V*_1_ and *V*_2_ are the titration volumes of the measured and blank samples (mL), respectively; *C* represents the actual concentration of HCl (M); and *M* is the weight of the oyster sample (g).

### 2.3. Determination of the Total Viable Counts (TVC)

The total bacteria count was determined in plate count agar using the inverted plate method with minor modifications [13]. A 0.5 g sample was homogenized with 5 mL of sterile normal saline for 1 min. The homogenized samples were then diluted with agar solution at ratios of 1:10, 1:100, and 1:1000, and placed upside down in a thermostatic incubator (Ningbo Saifu Experimental Instrument Co., Ltd., Ningbo, China) at 30 °C for 72 h. Microbiological data were converted to logarithmic colony-forming units (CFU/g).

### 2.4. Sensory Evaluation

The sensory evaluation of oysters was carried out according to the method described [14] with slight modifications. The trained panel of 10 members (consisting of 6 women and 4 men aged between 22 to 27 years) was chosen for this study. As shown in Table 1, sensory parameters included odor, tissue color, tissue texture, shell color, mantle color, and gill morphology, which were scored based on their intensity. The total score was collected after the evaluation, and a score below 18 was considered unacceptable.

### 2.5. Determination of Glycogen Content

The glycogen content was determined directly using the Merck Sigma-Aldrich glycogen test kit (Solarbio, Beijing Solarbio Science & Technology Co., Ltd., Beijing, China).

### 2.6. pH Measurement

The pH determination was performed [13] with minor modifications. One gram of oyster meat was homogenized with 10 mL of 20 mM sodium iodoacetate, and the pH was determined using a digital pH meter (ST3100, OHAUS, Ltd., Parsippany, NJ, USA).

### 2.7. Determination of ATP-Related Compounds

The measurement of ATP-related compounds was performed according to the method described previously [15]. A 2 g sample was ground with 10 times the volume of 5% perchloric acid, the supernatant was obtained after centrifugation at 6000× *g* for 3 min at 4 °C, and the pH was adjusted to 2.8–3.2 with KOH. The sample was centrifuged again to remove protein precipitate. The supernatant was adjusted pH to 7.0 using 0.5 M phosphate buffer and filtered through a 0.22 μm filter membrane. The filtrate was then stored at −20 °C for subsequent analysis using high-performance liquid chromatography (HPLC, Agilent 1260) with a column (Shodex GS-320 HQ, 7.5 mm I.D. × 300 mm).

The HPLC determination conditions were as follows: the mobile phase was 0.2 M phosphate buffer (pH 3.7), temperature of 30 °C, detection wavelength of 254 nm/4 nm, and flow rate of 0.6 mL/min.

To evaluate the freshness of oysters, K, K′ and AEC values were calculated according to the study reported [16].
*K value*% = (*HxR* + *Hx*)/(*ATP* + *ADP* + *AMP* + *IMP* + *HxR* + *Hx*) × 100%(2)
*K′ value*% *=* (*IMP* + *HxR* + *Hx*)/(*ATP* + *ADP* + *AMP* + *IMP* + *HxR* + *Hx*) × 100%(3)
*AEC value*% *=* 1/2 × (2*ATP* + *ADP*)/(*ATP* + *ADP* + *AMP*) × 100%(4)

*Abbreviations*: ATP, adenosine triphosphate; ADP, adenosine diphosphate; AMP, adenosine monophosphate; IMP, inosine monophosphate; HxR, inosine; Hx, hypoxanthine; AdR, adenosine.

### 2.8. Statistical Analysis

The data were statistically evaluated using two-way ANOVA with IBM SPSS Statistics 26.0 (IBM Corp., Armonk, NY, USA) and Duncan’s multiple range tests with significance at *p* < 0.05. Statistical values are expressed as mean ± standard deviation (SD). Furthermore, the heat maps were drawn using Origin software (version 22.0).

## 3. Results and Discussion

### 3.1. TVB-N Value

As an important indicator for evaluating the degree of aquatic product spoilage, a higher TVB-N value indicates a more serious degree of aquatic product spoilage [17]. As shown in Figure 1, the initial TVB-N values of the whole oyster in the enzymatic groups and the control group were 5.37 ± 0.15 and 5.85 ± 0.04 mg/100 g, respectively, which indicated the high freshness of the oyster. A previous study [18] reported that the initial TVB-N value in Pacific oysters (*Crassostrea gigas*) was 5.25 ± 0.14 mg/100 g, respectively. The TVB-N values in oysters exhibited a gradual increase, likely due to the activity of bacteria and endogenous enzymes, which led to the production of amine substances such as ammonia, methylamine, dimethylamine, trimethylamine, etc., with unpleasant odors [8]. However, after 7 d of storage, the WR group exhibited a higher level of TVB-N value compared to the control group, which could be attributed to the fact that the addition of the neutral protease could accelerate proteolysis in meat products. A similar observation was also found in grass carp (*Ctenopharyngodon idellus*) [19]. The TVB-N values of shucked oysters packed in 3.5% brine and stored at superchilled temperature increased at a slower rate than those stored at refrigerated temperature (Figure 1), probably because of the delayed growth of protease-producing bacteria and protease activity at low temperature [20]. The WS and DeS groups showed a slower increase in TVB-N values than the DS group, which could be attributed to the smaller temperature fluctuations, low drip losses, and high water-holding capacity caused by the high environmental humidity during storage [12]. In addition, high humidity could also dilute the enzyme concentration, thus slowing the rise of TVB-N values. A previous study [21] indicated that a high concentration of NaCl could reduce the levels of biogenic amines produced by bacterial protein degradation. For the WS (−1.5 °C) and WR (4 °C) groups, the effect of different storage temperatures on TVB-N values of oysters was significantly different, even with the same packing method. But in the case of the WS, DeS, and DS groups, the TVB-N value of oysters packaged with different storage media did not differ significantly in the first 5 d of storage at −1.5 °C. As the storage time continued to increase, the oysters soaked in 3.5% brine (WS group) achieved better freshness, as indicated by the lowest TVB-N value. The results showed that the WS, DeS, DS, WR, and control groups exceeded the allowable TVB-N value of oyster shelf life (10 mg/100 g) on the 10th, 10th, 7th, 5th, and 5th days, respectively. In summary, superchilling has a positive effect on the preservation of oysters and plays a dominant role among many factors, and the use of brine further enhances the maintenance of freshness.

The changes in TVB-N values of various parts of the oyster were also determined to explore the mechanisms of quality variation (Figure 1). The initial TVB-N values of the mantle, adductor muscle, gill, and body trunk of oysters were 2.52 ± 0.04, 2.80 ± 0.07, 3.10 ± 0.28, and 6.56 ± 0.60 mg/100 g, respectively. It has been suggested that the degradation of protein and nonprotein nitrogenous compounds contributed to the increase in TVB-N value [13]. Thus, the higher TVB-N value was measured in the body trunk due to its high content of protein and endogenous enzymes. The changes in TVB-N values in the four parts during storage were similar to those observed in the whole oyster, with values increasing exponentially. Therefore, the use of wet superchilling to preserve oysters showed that TVB-N values increased slowly both in all parts of the oyster and the whole sample.

### 3.2. The Total Viable Counts (TVC)

Microbial activity is one of the main factors contributing to the deterioration and off-flavor in oysters [22]. As shown in Figure 2a, the initial TVC values of the enzymatic group and the control group were 3.31 ± 0.17 lg CFU/g and 3.07 ± 0.08 lg CFU/g, respectively. A previous study [23] also reported that the initial TVC of shucked white scar oysters (*Crassostrea. belcheri*) was 3.7 lg CFU/g and increased dramatically during storage at 4 °C in 4% brine. During the first three days, the TVC of the WS group, DeS group, and DS group decreased. After five days, the TVC began to increase, with the lowest count observed in the WS group. This is probably because the 3.5% NaCl could inhibit the growth of microorganisms, such as *Vibrio parahaemolyticus* and *Vibrio vulnificus* [24]. Early bacterial stages of oyster strains vary in their tolerance to different conditions; some strains with poor low-temperature tolerance fail to grow or even die during early storage, resulting in a reduction in TVC. Therefore, the rise of TVB-N (Figure 1) during the first 5 d of storage in oysters was dominated by endogenous enzymes, and the subsequent increase was caused by a combination of endogenous enzymes and microbial activity. There was no significant difference in TVC between the WR and control group during the whole storage period, indicating that the addition of neutral protease did not affect the bacterial phase environment.

At storage time beyond 5 days, cryophilic strains such as fecal coliforms and *E. coli* continue to proliferate eventually, leading to increased TVC [25]. However, the TVC in all groups did not reach the upper acceptable limit of microorganisms in fresh oysters (5.7 lg CFU/g) throughout the whole study [26]. At the end of storage, the TVC values in the superchilling group (WS, DeS, and DS groups) were significantly lower than those in the refrigeration group, where the TVC exceeded 5 lg CFU/g. Similar patterns were reported in sturgeon (hybrid, *Acipenser schrenckiid* × *Huso dauricus*) and pork [27]. Thus, although microbial growth was inhibited at low temperatures and in brine, oysters continued to deteriorate during storage, mainly due to the action of endogenous enzymes.

### 3.3. Sensory Evaluation

Sensory evaluation was used to fully assess the effect of preservation method and temperature on oyster quality (Figure 2b). The fresh oysters were intact and had a strong characteristic appearance, taste, and odor; thus, the oysters on day 0 were classified as raw. In general, the sensory scores of all samples showed a continuous decrease with the storage time, indicating that the odor of oysters gradually changed from pleasant to putrid type as the storage time increased; as the tissues gradually softened and lost elasticity, the color became darker, showing more brown or yellow, and the gill also changed from clear to fuzzy. During the storage period, the sensory scores of the superchilling group were significantly higher than those of the refrigeration group. A similar observation was also reported in grass carp during fermentation [28]. Compared to the WR and control groups, although the neutral protease treatment reduced the shelf life of oysters, the brine treatment not only had a bactericidal effect to slow down oyster spoilage but also enhanced the flavor, hence the sensory scores of the WR group were higher than those of the control group. A sensory score greater than 18.00 is considered acceptable. The results showed that shucked oysters wrapped in 3.5% brine were acceptable for a maximum of 5 d in refrigeration and up to 10 d in superchilling conditions. A previous study [24] also recommended that the immersed white-scar oysters (*C. belcheri*) should not be consumed after 10 d of storage.

### 3.4. Glycogen Content

Glycogen, as a form of energy stored in oysters and other shellfish, could participate in the glycolytic pathway during shellfish preservation through acidification by hexokinase to produce a series of organic acids with consequent influence on the pH and flavor of the muscle [29]. As shown in Figure 3, the lowest glycogen content of 5.56 ± 0.27 mg/g in the control group was found in the mantle, followed by the body trunk (66.21 ± 15.03 mg/g), the adductor muscle (60.94 ± 14.70 mg/g), and the gill (20.53 ± 0.89 mg/g). After enzymatic hydrolysis, the glycogen content of all four parts increased. As storage time increased, the glycogen of WR and control groups decreased more significantly than those of the WS, DeS, and DS groups. The observed cell breakage and glycogen degradation in oysters may be attributed to the freeze–thaw phenomenon at −1.5 °C and relevant endogenous enzymes. Moreover, there was no significant difference in the glycogen content between the superchilling group and the refrigeration group on the same day. This suggests that the storage medium and the neutral protease did not affect oyster glycogen degradation.

### 3.5. pH

Microbial activity, glycolysis of glycogen, and decarboxylation or deamination of amino acids are closely related to changes in pH. Figure 4 shows that the initial pH values of the mantle, adductor muscle, gills, and body trunk of the control group were 6.53 ± 0.03, 6.71 ± 0.05, 6.45 ± 0.03, and 6.46 ± 0.07, respectively, which were similar to the previous results [13]. The WR group had a higher level of pH compared to the control group, which could be due to the addition of the neutral protease. The pH of all four tissues decreased throughout the storage period, with the fastest decline in the body trunk, followed by the gills, mantle, and adductor muscle. The decrease in pH might be attributed to the decomposition of glycogen and ATP-related compounds [30]. A previous study [31] suggested that oysters with a pH above 6.0 were considered to be of good quality. The pH of the WS, DeS, and DS groups ranged from 6.22 to 6.70 during storage, suggesting that the quality of the oyster was well maintained at −1.5 °C. It is worth noting that exposure to high concentrations of inorganic salts may affect the solubility of myofibrillar proteins and generate volatile bases, which could delay the pH decrease during storage [32]. For both the WR and control groups, the pH of oysters was below 6.0 after 7 d, which is lower than the general shelf life pH range for oysters specified in China (6.2~8.5), suggesting that the shelf life of oysters stored under this condition is only 5 d, with results consistent with those of TVB-N values (Figure 1). However, storage conditions had little significant impact on the change in pH in oysters on the same day.

### 3.6. ATP-Related Compounds

Nucleotide is a crucial indicator for assessing flavor and freshness of oyster meat. To understand the degradation mechanism of ATP-related compounds in postmortem oysters, the levels of ATP and its degradation metabolites were analyzed in all parts of the samples. Previous studies have confirmed two kinds of ATP degradation pathways in oysters (ATP→ADP→AMP→IMP/AdR→HxR→Hx) [33]. As shown in Figure 5, for the mantle of oysters, the ATP content of all samples initially increased and then gradually decreased during storage. The ADP content of each group showed fluctuations but no clear trend throughout the storage period, while the AMP content gradually increased. Only a small amount of IMP, HxR, Hx, and AdR could be detected during storage, indicating that two AMP decomposition pathways (IMP and AdR) existed simultaneously in the mantle. But for the adductor muscle, the initial amount of ATP is less than 0.3 μM/g, and the HxR and Hx concentrations were higher than in other tissues during storage, indicating that artificial shucking and enzymatic treatments induced rapid degradation of ATP. The ADP and AMP contents of each group decreased gradually with the increase in storage time, and the AMP of the control group was lower than that of the other groups in the first 3 d. During the initial 5 days, both the WR and DS groups had consistently higher IMP content compared to the control group. In addition, a small amount of AdR was still detected in the adductor muscle, indicating that there were two AMP decomposition pathways in the adductor muscle. In terms of the gills, changes in ATP, ADP, and IMP were similar to those of the mantle. IMP content in the gills, in general, was higher than in the mantle. The ATP-related compounds of all groups remained stable at −1.5 °C, and Hx and HxR did not exceed 0.3 μM/g throughout storage. For the body trunk, ATP levels in the WR and control groups increased slowly, then decreased rapidly, while AMP content increased gradually. Significantly higher concentrations of ADP and IMP were consistently found in the WR group compared to the control group, with the difference being significant between 5 d and 7 d. During the whole storage period at −1.5 °C, high concentrations were determined. The changes in ADP, IMP, Hx, and HxR of all groups were similar, with nonsignificant differences in the WS, DeS, and DS groups. Under hypoxia conditions, postmortem oysters could still activate glycogen phosphatase to maintain ATP production through glycogen catabolism [34], which could be explained by the high level of ATP that remained in the mantle, gill, and body trunk of the oyster.

### 3.7. Changes in K, K′, and AEC Values in Oyster Tissues during Storage

The chemical freshness indices (K, K′, and AEC value) of oysters were calculated based on the levels of ATP-related compounds (Figure 6). Previous studies reported that aquatic products with K values less than 20% are considered very fresh, less than 50% as moderately fresh, and more than 70% as spoiled [29]. In any case of our study, both K and K′ changed similarly. For the adductor muscle, the K values increased slowly in the WS group. The DeS group showed an overall upward trend, except for a slight decrease from day 5 to day 10. The K value of the control group increased rapidly and decreased slightly by the end of the storage period. For the mantle, gill, and body trunk, the K and K′ values of all samples remained below 20%. However, even when the K value was below 20%, the sensory evaluation results indicated that the oysters were not fresh. Therefore, the K-value was not suitable for shellfish evaluation. The AEC value of the adductor muscle tended to increase slowly in the early stage of storage under all treatments but did not change significantly throughout the storage stage, whereas the AEC. values of the WS, WR, and control groups gradually decreased with increasing storage time in the mantle, gills, and body trunk. However, the AEC values of the DeS and DS groups did not change significantly over time.

### 3.8. Relationships among Freshness Indicators of Oyster Tissues, TVB-N Value, and Sensory Score during Storage

The Pearson’s correlation coefficients with statistical significance among TVB-N value, sensory score, and freshness indices are shown in Figure 7a. The TVB-N value of oysters was significantly negatively correlated with the sensory score, AEC-Mantle, AEC-Gill, and AEC-Body trunk (*p* < 0.05), while it was positively correlated with K-Mantle, K-Adductor muscle, K-Gill, K′-Mantle, K′-Adductor muscle, K′-gill, and AEC-Adductor muscle (*p* < 0.05). However, the applicability of some freshness evaluation indexes is affected by the packaging medium. Under wet storage (Figure 7b) conditions, TVB-N values were also positively correlated with K-body trunk (*p* < 0.05) and K′-body trunk (*p* < 0.05). Under dry dew conditions (Figure 7c), the correlation between TVB-N values and K-Mantle was not significant. Under dry storage conditions (Figure 7d), some freshness indices (K-Gill, K-Body trunk, K′-Body trunk, AEC-Gill) did not show a high correlation with TVB-N values and sensory scores (Figure 7d). These results suggest that K-Adductor muscle, K′-Mantle, K′-Adductor muscle, K′-Gill, AEC-Mantle, and AEC-Body trunk could be used as the most useful freshness indices for postmortem oysters.

## 4. Conclusions

The results of this study suggest that pretreatment of oysters with neutral protease can not only effectively enhance the initial nucleotide and glycogen content in edible parts but can also cause a large number of flavor nucleotides (such as IMP) accumulation in the body trunk of oysters, resulting in a better flavor. The use of a 3.5% brine wet or dry exposure storage method could slow down the decline in TVB-N and pH value, with certain sterilization functions, effectively maintaining the freshness of oysters at −1.5 °C. Superchilling wet or dry exposure storage was the most effective method for extending the shelf life of postmortem oysters to 10 d, as compared to 5 d of refrigeration, while maintaining the freshness quality characteristics. In addition, the changes in biochemical parameters in different parts of the oyster varied considerably. However, pretreating the oysters with proteolytic enzymes did not cause an accumulation of flavor nucleotides such as IMP. The K value, K′ value, and AEC value, which serve as the evaluation indexes of oyster freshness, were affected by the storage medium. Therefore, the focus of our future research will be on the accumulation of flavor nucleotides in oysters and the development of rapid, nondestructive methods for evaluating freshness.

## Figures and Tables

**Figure 1 foods-13-01273-f001:**
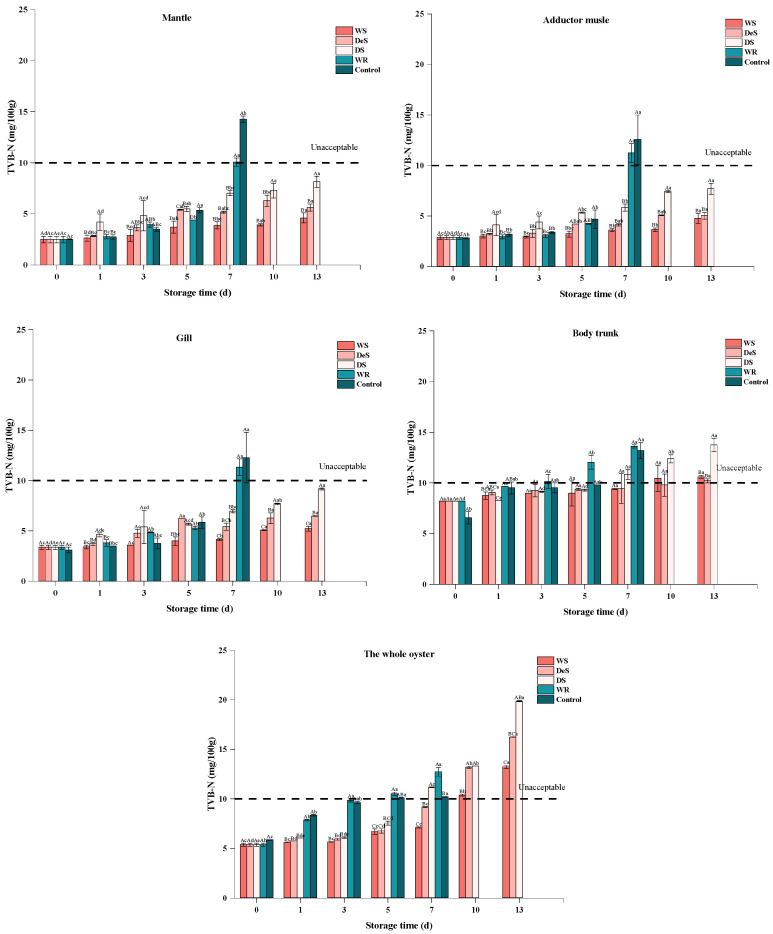
Changes in TVB-N in various tissues and the whole oyster during storage. Lowercase letters indicate significant differences (*p* < 0.05) at the same groups of different storage times, and capital letters indicate significant differences (*p* < 0.05) for different groups at the same storage time.

**Figure 2 foods-13-01273-f002:**
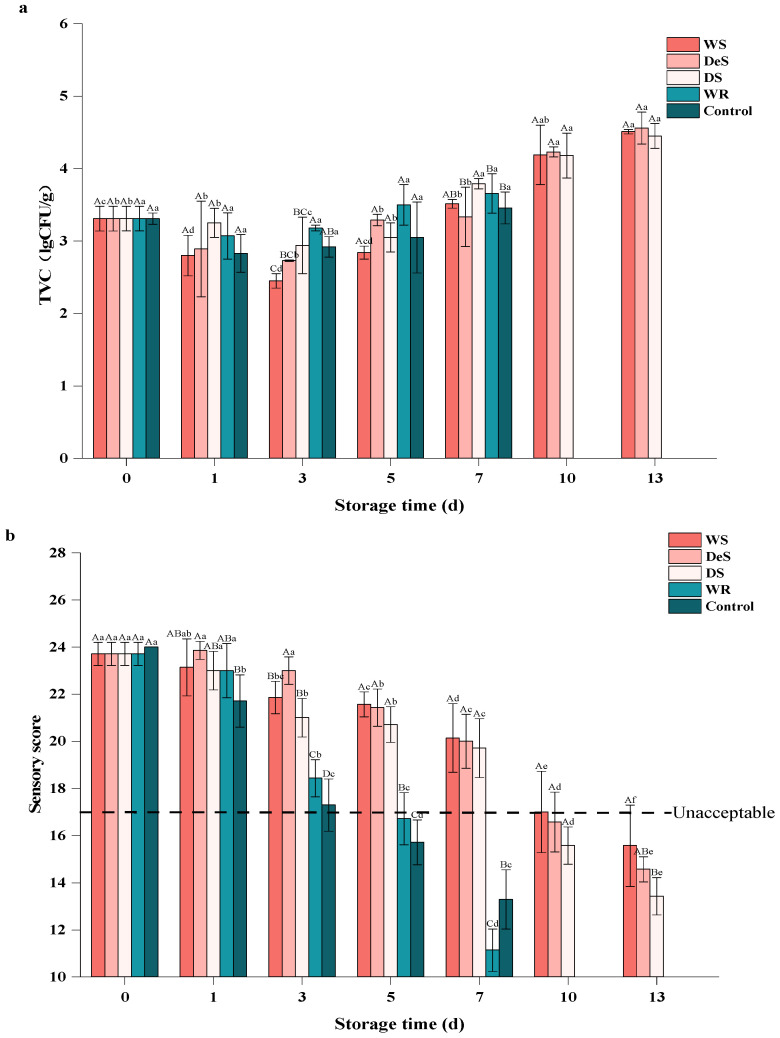
Changes in TVC (**a**) and sensory score (**b**) of oyster during storage. Lowercase letters indicate significant differences (*p* < 0.05) at the same groups of different storage times, and capital letters indicate significant differences (*p* < 0.05) for different groups at the same storage time.

**Figure 3 foods-13-01273-f003:**
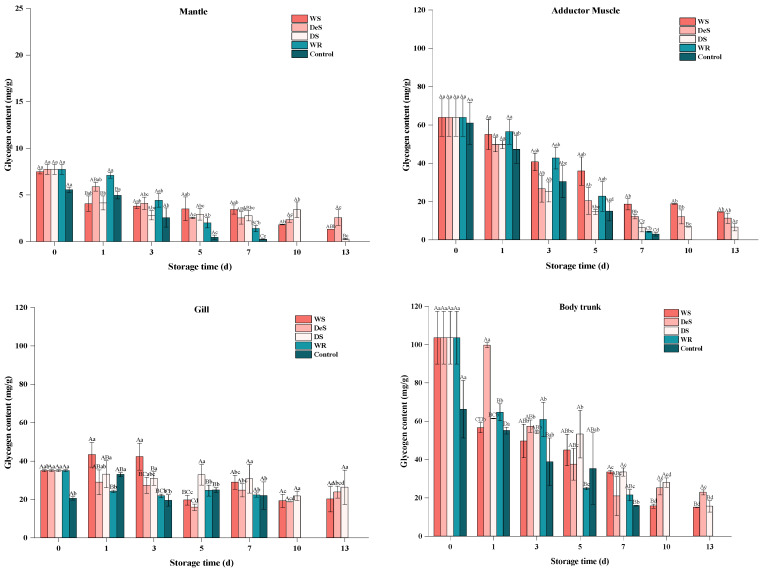
Changes in glycogen in various tissues of oyster during storage. Lowercase letters indicate significant differences (*p* < 0.05) at the same groups of different storage times, and capital letters indicate significant differences (*p* < 0.05) for different groups at the same storage time.

**Figure 4 foods-13-01273-f004:**
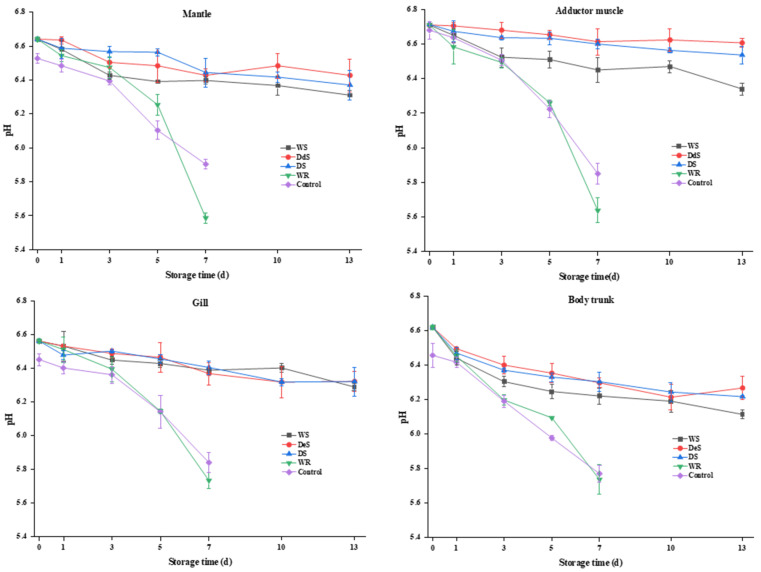
Changes in pH in various tissues of oyster during storage.

**Figure 5 foods-13-01273-f005:**
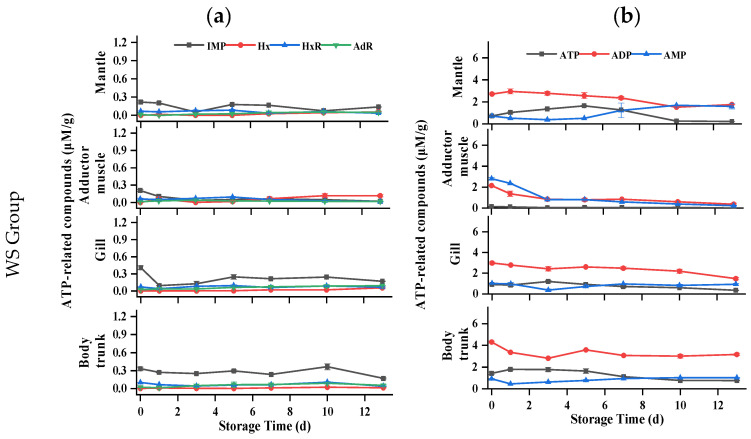
Changes in ATP-related compounds in various tissues of oyster during storage. (**a**) IMP, Hx, HxR, and AdR in WS, DeS, DS, WR, and control groups; (**b**) ATP, ADP, and AMP in WS, DeS, DS, WR, and control groups.

**Figure 6 foods-13-01273-f006:**
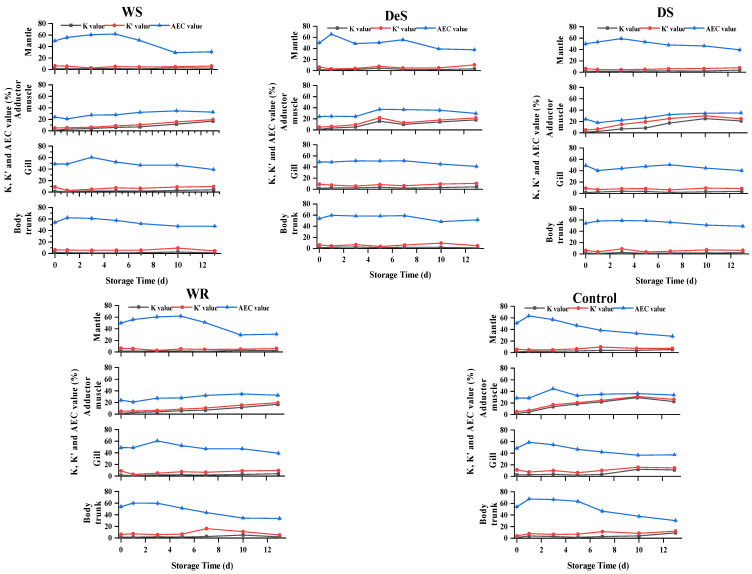
Changes in K, K′, and AEC value in various tissues of oyster during storage.

**Figure 7 foods-13-01273-f007:**
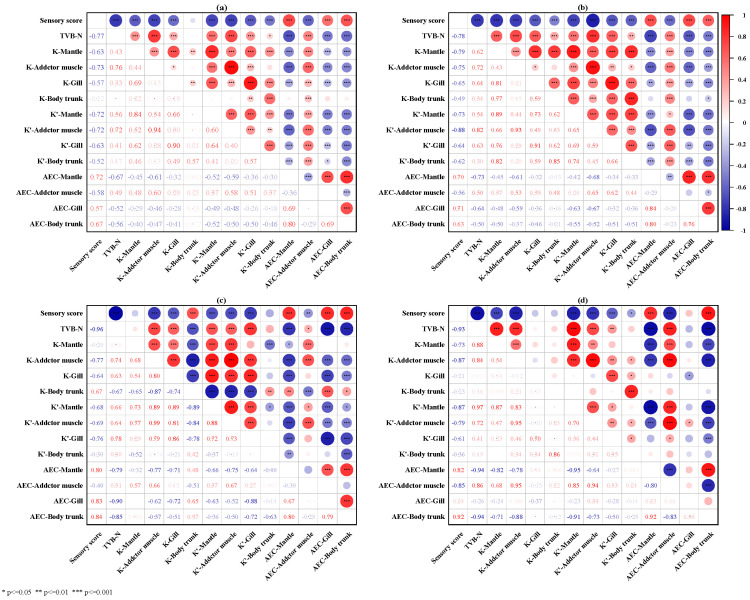
The correlation heatmap among TVB-N value, sensory score, and freshness indices under different storage conditions: (**a**) under all storage conditions; (**b**) wet storage; (**c**) dry dew storage; (**d**) dry storage. The red and blue colors in the graph represent positive and negative correlations, respectively, and the depth of the color represents the intensity of the correlation. * *p* ≤ 0.05, ** *p* ≤ 0.01, *** *p* ≤ 0.001.

**Table 1 foods-13-01273-t001:** Freshness guide for oysters.

Parameters	Score			
	4	3	2	1
Odor	Strong inherent odor	Light inherent odor	Light putrefaction	Putrefaction
Body color	Cream white	White	Tawny/beige	Yellow/light brown
Texture	Firm and elastic	Soft and less elastic	Slightly mushy	Mushy
Adductor	Pale white Translucent	Light gray Translucent	Light gray Partially opaque	White Opaque
Mantle	Brown/Black	Slight fading	Mostly faded	Faded
Gill	Gill filaments Well defined	Filaments Less defined	Filaments Poorly defined	Filaments Undefined

## Data Availability

The original contributions presented in the study are included in the article, further inquiries can be directed to the corresponding author.

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
