# Peer review of "Effect of Neutral Protease on Freshness Quality of Shucked Pacific Oysters at Different Storage Conditions"

_foods, 2024, doi:10.3390/foods13081273_

Round 1
Reviewer 1 Report
Comments and Suggestions for Authors
Dear authors, the revised manuscript is interesting, however the following comments need to be addressed:
Introduction section
Line 32: remove space… (pre-treated
Line 45,56,62: shelf life or shelf-life like in line 23… homogenize terms through the manuscript
Materials and methods section
Note: include the origin and purity of the chemicals and reagents used
Line 74: rewrite… (123.24 ± 9.74 g), were
Line 86: rewrote… -1.5 ± 0.2°C
Line 86: 4.0 ± 0.2°C
Line 94: rewrite… as described previously [2].
Line 95: insert temperature used during centrifugation process.
Line 95: What were the conditions to homogenize the samples, rpm? time?
Line 95: insert equipment information, mixer (model, trademark, country)
Line 95: insert equipment information, centrifuge (model, trademark, country)
Line 96: respect MgO solution… What concentration was used?
Line 111: rewrite… according to the method described [14] with
Line 122: rewrite… was performed [13] with minor
Line 127: rewrite… described previously [15]. A 2 g
Line 135: insert space… temperature, 254
Line 138: rewrite… reported [16].
Line 146: In studies where control and n number of treatments are compared, and you are measuring n number of response variables over storage time (n days...), it would be most appropriate to use a two-way ANOVA. The reason is that you have two main sources of variation that you are evaluating: the different treatments and the effect of storage time. So, you need an ANOVA that can simultaneously analyze the influence of both factors on the response variable. Factor one would be the type of treatment (control and n number of different treatments), and factor two would be the storage time (n days...). This will allow you to determine if there are significant differences between treatments, if there are significant differences over time, and if there are any significant interactions between treatments and storage time. Therefore, a two-way ANOVA would be the most appropriate option to evaluate the results in this scenario. It is necessary to use this type of ANOVA, and adjust the wording of the text based on the effect of both factors and double interaction.
Line 155: insert space… 5.37 ± 0.15 and 5.85 ± 0.04
Line 156: rewrite… A previous study reported that the initial TVB-N….. (Note: According to the author guide, the use of authors' names or surnames in the text should be avoided; only the corresponding reference number should be placed in square brackets).
Line 163: remove space… products. A
Line 181: is that 10th, 10th, 7th, 5 d and 5 d written, correct?
Line 189,190: insert spaces between number and ± symbol… Note: correct through the manuscript
Line 190: rewrite… It has been suggested
Line 197,198: missed text?
Line 202: did you mean log instead lg?
Line 222: According to the author guide, the use of authors' names or surnames in the text should be avoided; only the corresponding reference number should be placed in square brackets (Note: correct through the manuscript)
Line 207: use the non-abbreviated form of scientific names when they appear for the first time in text
Line 218: It is necessary to place figure 2b in the section of the paragraph that corresponds to it
Line 229,230,268,269: missed text?
Line 279: ATP-related
Line 373: occasionally days or d appears in the text, it is necessary to homogenize terms
Line 393: use the correct text format for the scientific name
Line 393: use the abbreviated format for the journal name
Note: Carefully review the format of each of the references in this section
Note: use text formats (size and font) indicated in the authors' guide. You can consult the Microsoft word template of the journal
Author Response
Detailed information on the revisions could be found in the attachment: revision note 1.

Reviewer 2 Report
Comments and Suggestions for Authors
The manuscript deals with the effect of neutral protease on freshness quality of shucked pacific oysters at different storage conditions.
Please revise the manuscript to reduce the level of similarity (26%).
The English language must be revised.
Please separate values from units, e.g. “-1,5 ºC” not “-1,5ºC”.
Abstract
This section is vague. Please present your main results.
Materials and methods
“The sensory evaluation of oysters was carried out according to the procedure of Chen et al [14] with slight modification. The trained panel of 10 members was chosen for this study. As shown in Table 1, sensory parameters included odor, tissue color, tissue texture, shell color, mantle color, and gill morphology which were scored based on the intensity. The total score was collected after the evaluation, and a score below 18 was considered unacceptable.”??how many men??women??ages??
Color analysis??
Texture analysis??
Results and discussion
Pictures of each sample??
“As an important indicator for evaluating the degree of aquatic product spoilage, the higher TVB-N value indicates the more serious degree of aquatic product spoilage [17]. As shown in Figure 1, the initial TVB-N values of the whole oyster in the enzymatic groups and the control group were 5.37± 0.15 and 5.85± 0.04 mg/100 g which indicated the high freshness of the oyster.”??Figure 1, please add a line for the TVB-N maximum value allowed for human consumption.
Figures 5 and 6 are confusing. Please revise accordingly.
References
Please format the scientific names in italic.
Comments on the Quality of English LanguageMinor editing of English language required.
Author Response
Detailed information on the revisions could be found in the attachment: revision note 2.

Round 2
Reviewer 1 Report
Comments and Suggestions for Authors
It is recommended to use the correct text format in the references section, you can review the authors' guide or the information contained in the Microsoft word template
Author Response
Response to Reviewers
We would like to thank the reviewers for carefully reading our manuscript. We appreciate the comments and suggestions. In the following, we include a point-by-point response to the comments from each reviewer. In the revised manuscript, all the changes have been highlighted in red.
Replies to the reviewer 1:
[Comment 1] It is recommended to use the correct text format in the references section, you can review the authors' guide or the information contained in the Microsoft word template
Our Response: We appreciate your comments, we carefully checked all the references to make sure they were cited correctly. In addition, we corrected all reference citation formats using ACS style based on content in the guide for authors of Foods.

Reviewer 2 Report
Comments and Suggestions for Authors
Figures 5 and 6 are still confusing. Please revise accordingly, e.g. change charts to portrait alignment.
Comments on the Quality of English Language
Minor editing of English language required.
Author Response
Response to Reviewers
We would like to thank the reviewers for carefully reading our manuscript. We appreciate the comments and suggestions. In the following, we include a point-by-point response to the comments from each reviewer. In the revised manuscript, all the changes have been highlighted in red.
Replies to the reviewer 2:
[Comment 1] Figures 5 and 6 are still confusing. Please revise accordingly, e.g. change charts to portrait alignment.
Our Response: Thanks for your suggestion. We have corrected Figures 5 and 6 to ensure that their content is clear. As shown on page 11-13 of the manuscript.
[Comment 2] Minor editing of English language required.
Our Response: Thanks for your suggestion. We invited a professor who studied in Canada to help us revise the grammar, and the corrections are marked in red.
